# Design and Optimization of Ram Air–Based Thermal Management Systems for Hybrid-Electric Aircraft

Hagen Kellermann [1,*], Michael Lüdemann [1], Markus Pohl [2] and Mirko Hornung [1]

1   Bauhaus Luftfahrt e. V., Willy-Messerschmitt Straße 1, 82024 Taufkirchen, Germany
2   Institute of Jet Propulsion and Turbomachinery, RWTH Aachen University, 52062 Aachen, Germany
*   Correspondence: hagen.kellermann@bauhaus-luftfahrt.net

**Abstract:** Ram air–based thermal management systems (TMS) are investigated herein for the cooling of future hybrid-electric aircraft. The developed TMS model consists of all components required to estimate the impacts of mass, drag, and fuel burn on the aircraft, including heat exchangers, coldplates, ducts, pumps, and fans. To gain a better understanding of the TMS, one- and multi-dimensional system sensitivity analyses were conducted. The observations were used to aid with the numerical optimization of a ram air–based TMS towards the minimum fuel burn of a 180-passenger short-range partial-turboelectric aircraft with a power split of up to 30% electric power. The TMS was designed for the conditions at the top of the climb. For an aircraft with the maximum power split, the additional fuel burn caused by the TMS is 0.19%. Conditions occurring at a hot-day takeoff represent the most challenging off-design conditions for TMS. Steady-state cooling of all electric components with the designed TMS is possible during a hot-day takeoff if a small puller fan is utilized. Omitting the puller fan and instead oversizing the TMS is an alternative, but the fuel burn increase on aircraft level grows to 0.29%.

**Keywords:** thermal management; hybrid-electric aircraft; ram air–based cooling; compact heat exchangers; meredith effect

## 1. Introduction

The introduction of (hybrid-)electric powertrains to future aircraft is one of the innovations that could help to achieve the ambitious goal of a 75% reduction in $CO_2$ emissions by the year 2050 set by the European Commission's Strategic Research and Innovation Agenda [1]. Thermal management is one of the key challenges for the successful realization of such powertrains [2].

Thermal management systems (TMS) were already part of early motorized aircraft, especially for the cooling of piston engines. When the engine power density increased, air cooling became insufficient and additional radiators were installed to reject heat from the oil system to ambiance. The Mustang P-51D and Messerschmitt Bf 109 are examples of aircraft which had these radiators installed inside a duct with a diffuser and a nozzle to reduce cooling air drag utilizing the so-called Meridith effect [3]. This principal architecture of a ram air–based cooling system is still present in modern aircraft systems, e.g., in the environmental control system [4].

With the introduction of gas turbines, and for turbofan engines especially, engine thermal management became a less critical issue for commercial aircraft because of the large, steady airflow that carries most of the engine's waste heat to ambiance. However, the continuous increase in turbine entry temperature and the introduction and further development of new technologies—for example, a gearbox for geared turbofan engines—have led to increased heat loads in modern aircraft engines. A summary of the development of engine waste heat and corresponding TMS developments can be found in [5].

Over the last two decades, research in (hybrid-)electric powertrains as an alternative to gas turbines has significantly increased. One of the key challenges for both realizing

a theoretical benefit on aircraft level and successfully implementing first demonstrations is the thermal management of up to multi-megawatt electric powertrains [6,7]. Besides the high efficiency of electric components compared to gas turbines, they have no natural large heat rejection system such as the engine exhaust, so only small amounts of heat can be dissipated naturally via conduction through the structure. Therefore, the TMS has to manage their entire heat load. Additionally, electric components typically have low operating temperatures compared to combustion engines, which result in only small available temperature differences to ambient conditions for the TMS.

In recent research on hybrid-electric aircraft (HEA), the TMS is addressed more frequently and with increasing level of detail. For the NASA STARC-ABL concept, a specific power of 0.68 kW/kg of the TMS was assumed [8]. A hybrid version of the NASA N+4 Refined SUGAR research platform was designed with a dynamic model of a TMS for both the electric system and the engine oil system. The system was designed for conditions during a hot-day takeoff (HDTO), which, together with a low allowable battery temperature, resulted in a ram air cooler of about 150 kg. However, a 50% mass reduction was shown for an increase in battery temperature of 20 °F [9]. Further analysis of the concept, including various off-design points, showed an increase in design mission fuel burn ($FB$) of 3.4% due to TMS mass, power, and drag [10]. With additional optimization, such as decoupling the battery cooling loop, the $FB$ increase was reduced to 0.75% [11]. In [12], a ram air–based TMS was designed for a vertical takeoff and landing (VTOL) vehicle with steady-state and transient methods. Sensitivities of key parameters of the developed compact heat exchanger (HEX) model were shown, as were Pareto fronts for a system optimization towards minimum system mass and power required by a puller fan. The final TMS of the VTOL had a mass of 171.63 kg and required 257.6 kW of power.

For HEA, the potential of using existing aircraft surfaces as alternative heat sinks was investigated, resulting in an indication that smaller aircraft can reject large parts of their heat load via the skin [13]. In a more detailed investigation, a TMS utilizing recirculating fuel underneath the wing surfaces for cooling of a 180-passenger short-range HEA was designed [14]. Despite the promising results, these surface cooling concepts have major disadvantages, such as the low available cooling power at low flight velocities and the low amount of coolant in case of fuel cooling towards the end of the mission. Therefore, a ram air–based TMS was considered for this study.

The research on ram air–based TMS has already developed some sensitivities and optimization for the compact HEX rather than solely solving the thermal management issue of one specific HEA. In this study, an even broader approach was chosen. The objective was threefold: Firstly, a static model of all necessary components for a ram air–based TMS was developed. Secondly, the overall system sensitivities were studied rather than just those of the compact HEX. Thirdly, different TMS architectures were optimized towards a weighted objective function derived from a 180-passenger short-range partial-turboelectric aircraft. The study will further improve knowledge of ram air–based TMS and their impacts on HEA. It will thereby enable future studies on HEA to assess their performances in more detail.

## 2. Models and Methods

The following section describes all required component models of the ram air–based TMS. At the end of this section, the partial-turboelectric aircraft and the derivation of its $FB$ sensitivities for later use as an objective function are presented.

Figure 1 shows an exemplary centralized TMS architecture with all electric components being cooled in parallel. It requires the following components:

1. Coldplates to receive heat from the electric components and transfer it to the coolant.
2. A compact HEX to reject the collected heat to ambiance.
3. A diffuser to reduce cooling air speed and thereby the cold-side pressure loss of the compact HEX.
4. Optionally, a puller fan to increase cooling air flow.

5. A nozzle to recover some of the momentum of the cooling air and thereby reduce drag.
6. Pipes to transfer the coolant.
7. A pump to recover the pressure loss of the coolant.

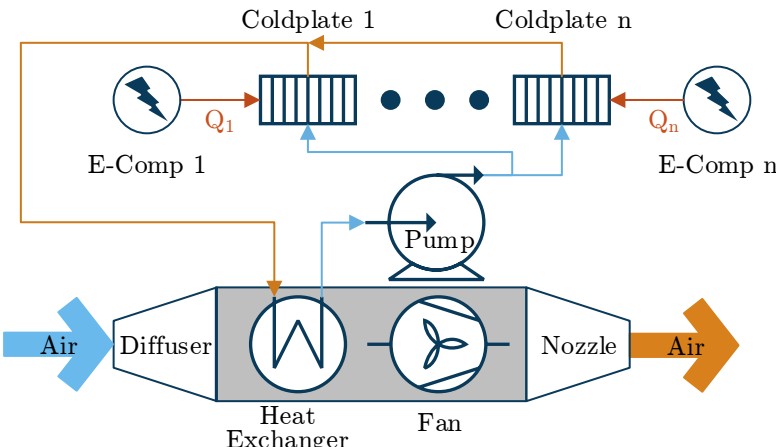

**Figure 1.** Centralized parallel thermal management system (TMS).

*2.1. Coldplates*

Coldplates are flat components with internal liquid flow to cool electronic devices, such as chips. Research trends towards lower thermal resistances ($R_{th}$) of future coldplates—for example, by decreasing the hydraulic diameters ($d_H$) of microchannels or by integrating the cooling channels closer to the working parts of the electronics [15]. Here, a simplified model of a coldplate is used not only for the cooling of the power electronics but also as a substitute for a model of the internal cooling of electrical machines. Despite the inlet properties (pressure ($p$), temperature ($T$), and heat load ($Q$)), the model only requires thermal insulance ($r_{th}$), maximum junction temperature ($T_{cp}$), area density ($\rho_A$), and design pressure loss ($\Delta p_{des}$) as inputs. These can be estimated from existing manufacturer data or research articles for future coldplate technology. The off-design performance is analytically derived, assuming straight parallel microchannels with laminar flow. A detailed explanation of the implemented coldplate model is provided in Appendix A.1. To validate the model, data from a numerical study of a microchannel coldplate is used [16]. The design point of the model was set to the highest Reynolds number ($Re$), and for the off-design performance, the mass flow rate ($w$) was subsequently decreased. All inputs to the design model are listed in Table A3 in Appendix A.2. The results of the validation are shown in Figure 2.

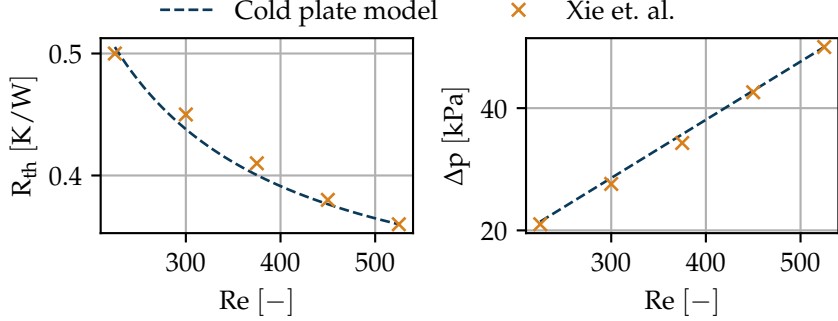

**Figure 2.** Coldplate model validation for thermal resistance (**left**) and pressure loss (**right**) with data from [16].

The predicted performances for both parameters ($R_{th}$ and $\Delta p$) are within 2% of the validation data. The slight inaccuracy stems from the errors made in the visual acquisition

of the data and the simplifications of the model. For the use in preliminary aircraft design, the accuracy is acceptable.

## 2.2. Compact Heat Exchanger

Heat exchangers can be built in many different architectures that have been described and categorized by different authors, e.g., [17,18]. Models attempting to cover all the different HEX types are therefore limited to a very low level of detail, which is not sufficient for the aim of this study to predict mass, dimensions, power, and drag of the TMS. However, due to the specific requirements of aircraft, only light, compact HEXs are considered. In [19], the most promising types of HEXs for aircraft applications are summarized as plate-fin heat exchangers (PFHE), printed circuit heat exchangers (PCHE), and in the future, microchannel heat exchangers.

There is no hard distinction between these three, as PFHE is a description of the overall architecture (plates and fins), PCHE is a description of the manufacturing technique (additive), and microchannel is a description of the layout on the microscopic level. Therefore, a HEX could match all three categories if it is an additively manufactured PFHE with very small channels. Thus, from a modeling perspective, it is only one type, which can be described as a single-phase, multi-pass, cross-flow HEX in overall counterflow arrangement. Both design and performance calculations were derived from the detailed procedures described in [17] for PFHE. Adaptions for the number of transfer units (NTU), the effectiveness ($\epsilon$), and the dimensions of the HEX for multipass arrangements were implemented from [18]. The key equation for core mass velocity (*cmv*) from [17] then becomes:

$$
cmv_{des} = \sqrt{2\Delta p_{des}} \cdot \left[ \frac{f_{corr}}{j} \frac{ntu}{\eta_o} \cdot Pr^{\frac{2}{3}} \cdot \frac{1}{\rho_m} + 2 \cdot \left( \frac{1}{\rho_o} - \frac{1}{\rho_i} \right) + (1 - \sigma^2 + K_c) \cdot \frac{n_p}{\rho_i} \right.
$$
$$
\left. - (1 - \sigma^2 - K_e) \cdot \frac{n_p}{\rho_o} + (n_p - 1) \cdot K_{bt} \cdot \frac{\sigma^2}{\rho_m} \right]^{-0.5}
\tag{1}
$$

with corrected friction factor ($f_{corr}$), number of transfer units on one side (*ntu*), overall fin efficiency ($\eta_o$), Prandtl number (*Pr*), inlet, outlet, and mean density ($\rho_i$, $\rho_o$, and $\rho_m$), ratio of free flow to frontal area ($\sigma$), inlet, outlet, and bend loss coefficient ($K_c$, $K_e$, and $K_{bt}$), and number of passes ($n_p$).

The described algorithm can work with any HEX core as long as the parameters in Table 1 are given. The Colburn factor (*j*) and the Fanning friction factor (*f*) depend on *Re*, which means a correlation rather than one value has to be given. All other parameters are geometric and do not change in off-design operation. Three options for the HEX core are considered:

1.  Rectangular microchannels.
2.  Offset-strip fins.
3.  Louvered fins.

A detailed explanation for the calculation of all parameters in Table 1 for all three types of HEX core can be found in Appendix B.

## 2.3. Diffuser, Nozzle, and Pipes

In many TMS models, e.g., the model presented in [12], the diffuser pressure loss is assumed to be constant. However, at low flight speeds, this simple assumption may overestimate the actual pressure loss and lead to the necessity of a puller fan. Its installation should be carefully considered because it usually is less efficient than the main propulsion devices. Therefore, in this study, a Mach number (*Ma*) dependent pressure loss model is used for the diffuser.

A drawing of the two-dimensional diffuser model is shown in Figure 3. It has a rectangular cross section, an opening angle ($\theta$) in the z-direction, and a constant width (y-direction). Depending on the flight conditions, there is a pre-entry compression or

expansion, i.e., $A_0 \neq A_1$. The changes in fluid properties between the flow cross sections $A_0$ and $A_1$ are calculated with the isentropic relations. Inside the diffuser, the ideal pressure recovery factor ($c_p^*$) can be obtained from correlations found in [20]:

$$c_p^* = g_1 \cdot g_2 \cdot \left\{ 1 - \frac{1.03 \cdot (1-B)^2}{\overline{A_R}^2 \cdot \left[1 - 0.82 \cdot \overline{A_R}^{-0.07} \cdot B^{1/(2 \cdot \overline{A_R}-1)}\right]^2} \right\} \tag{2}$$

$g_1$ is a term depending on $Ma$ and diffuser area ratio ($A_R = A_2/A_1$), and $g_2$ is a term depending on $Re$ and relative inlet blockage ($B$). $\overline{A_R}$ is a corrected $A_R$ to account for the influence of the aspect ratio of the inlet cross section. Using $c_p^*$ implies a diffuser with optimal $\theta$, which for the 2-D diffusers is around $8°$. The outlet pressure is:

$$p_{2,s} = c_p^* \cdot \rho_1 \cdot v_1^2 + p_{1,s} \tag{3}$$

If $A_0 < A_1$, some air is spilled around the inlet and spillage drag occurs. It can be calculated according to [21,22]:

$$D_{spill} = K_{spill} \cdot [w_1 \cdot (v_1 - v_0) + A_1 \cdot (p_1 - p_0)] \tag{4}$$

$K_{spill}$ is an empirical coefficient accounting for the lip suction effect. $D_{spill}$ is added to the internal drag calculated from conservation of momentum equations over the entire system, i.e., from diffuser inlet to nozzle outlet.

**Table 1.** Required heat exchanger core parameters.

| Name | Symbol | Unit |
|---|---|---|
| Colburn factor | $j$ | − |
| Fanning friction factor | $f$ | − |
| Hydraulic diameter | $d_H$ | m |
| Plate space | $b$ | m |
| Area density | $\beta$ | $m^2/m^3$ |
| Fin thickness | $\delta$ | m |
| Fin thermal conductivity | $\lambda_f$ | $W/(m\,K)$ |
| Ratio finned to total heat transfer area | $A_f/A$ | − |

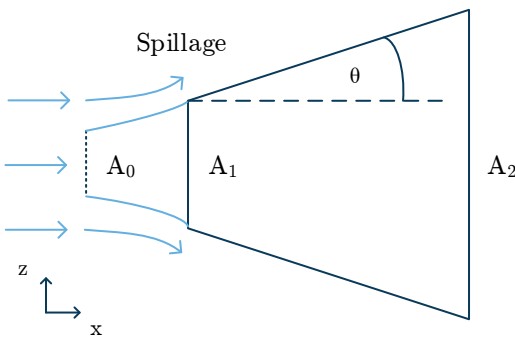

**Figure 3.** Diffuser model.

Since the nozzle has a negative static pressure gradient in the flow direction, its total pressure loss is less sensitive to shape and flow conditions than the diffuser. However, for the same reasons as mentioned above, it is important to have a pressure loss correlation sensitive to flow velocity rather than just a constant. It can be calculated according to [23]:

$$\Delta p_t = K_{loss} \cdot p_{1,t} \cdot \left[1 - \frac{p_{1,s}}{p_{1,t}}\right] \tag{5}$$

with shape-specific loss coefficient ($K_{loss}$) from [23]. Otherwise, the nozzle model uses area ratios to calculate outlet velocity and isentropic relations for the outlet fluid properties.

The pipe is modeled as a straight circular channel, and the well-known head loss formulas, e.g., from [24], are used to estimate pressure loss. For turbulent flow, the correlation from [25] is used to predict the friction factor.

All three models have simple geometric models to estimate their dry masses. In case of the pipe, a wet mass depending on the coolant is also available.

### 2.4. Pump and Fan

The puller fan is modeled as a repetition stage according to [22], i.e., the outlet velocity equals the inlet velocity. Isentropic relations are used to calculate the outlet fluid properties and compression work.

The pump model is simpler as the fluid is considered to be incompressible. Two efficiencies are implemented: The hydraulic efficiency ($\eta_{hyd}$) and the electric efficiency ($\eta_{elec}$). Mechanical power and outlet temperature are calculated as:

$$P_{mech} = \frac{\Delta p \cdot w}{\rho \cdot \eta_{hyd}} \tag{6}$$

$$T_2 = T_1 + P_{mech} \cdot \frac{1 - \eta_{hyd}}{c_v \cdot w} \tag{7}$$

### 2.5. Aircraft Fuel Burn Sensitivities

The aircraft used for the TMS design and optimization is designed to carry 180 passengers over a range of 1300 NM at a cruise speed of $Ma = 0.68$ (initial cruise altitude: 35,000 ft) and features a partial-turboelectric propulsion system.

The propulsion system is composed of advanced turboprop engines and turboelectrically driven wingtip propellers (WTPs). A key variable of this propulsion system architecture is the power split ($S_P$), which is defined as:

$$S_P = \frac{P_{WTP}}{P_{MP} + P_{WTP}} \tag{8}$$

where $P_{WTP}$ is the shaft power of the WTP and $P_{MP}$ is the shaft power of the turboprop engine's main propeller (MP). The design power of the electric system is determined by $S_P$ and $P_{MP}$ at the top of climb (TOC) of the aircraft design mission. This electric power remains constant unless the power of the gas turbine is lower than its TOC power of the design mission. In this case, $P_{WTP}$ is lowered accordingly to match the desired $S_P$. Further details about the propulsion system and aircraft are provided in [26]. The aircraft investigated in [26] and this study is visualized in Figure 4.

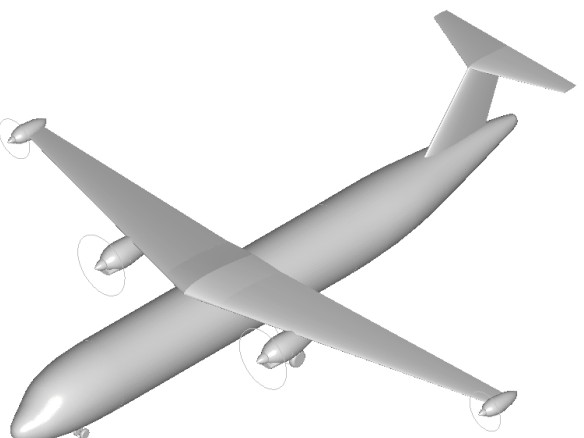

**Figure 4.** Aircraft design for $S_P = 30\%$.

To achieve an optimized TMS design on aircraft level, the impact of the variation of its most important parameters on an aircraft objective optimization variable is required. For this purpose, the sensitivity of the partial-turboelectric aircraft's *FB* (block fuel) to varying mass and drag increments due to the TMS integration was derived for three $S_P$ values (10%, 20%, and 30%). Regarding the additional mass of a TMS ($m_{TMS}$), the operating empty mass (OEM) was gradually increased to include an assumed $m_{TMS}$ of up to 1000 kg. In the same manner, the wing profile drag was increased to include an assumed TMS drag ($D_{TMS}$) of up to 1000 N since an integration into the wing was found to be reasonable. Consequently, every combination of $m_{TMS}$ and $D_{TMS}$ represents a new aircraft design. The resulting aircraft *FB* sensitivities for the three $S_P$ variations are similar in their relative *FB* changes ($\Delta FB$ values) to the *FB* of the respective baseline aircraft design. An exemplary *FB* sensitivity is presented in Figure 5.

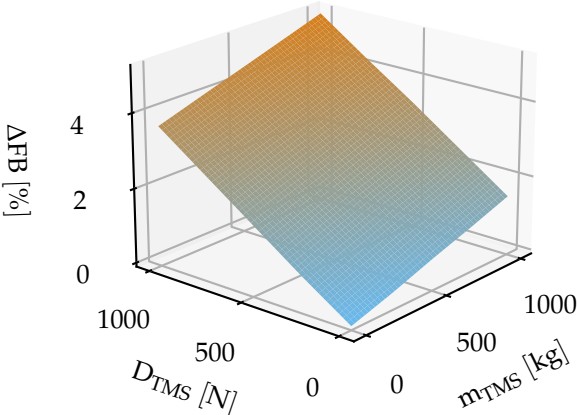

**Figure 5.** Aircraft *FB* sensitivity for $S_P = 30\%$.

Starting from the baseline aircraft design for $S_P = 30\%$, Figure 5 shows an increase in $\Delta FB$ of approximately 1.5% for a $m_{TMS}$ increment of 1000 kg and approximately 3.6% if $D_{TMS}$ is increased by 1000 N. These *FB* gradients of $\Delta m_{TMS}$ and $\Delta D_{TMS}$ are almost independent of each other, which leads to a sensitivity plane with only minimal curvature.

## 3. System Sensitivity Analysis

The following section investigates an aircraft *FB* sensitivity to all relevant parameters of the system. It establishes a general understanding of the system and verifies the implementation of the models. Additionally, computational costs in the following optimization (see Section 4) are reduced when parameters with low sensitivity can be set to a constant value. The sensitivity analysis is conducted at TOC conditions. However, HDTO conditions are more challenging for the TMS and are considered later in Section 4.2. $S_P = 30\%$ is used for the sensitivity analysis. The trends shown in this section are also valid for the other $S_P$ values. The heat loads of the design and the off-design point are shown in Figure 6.

Power electronics include inverters, rectifiers, and protection switches. The absolute values are rather close due to the aforementioned strategy of keeping the electric power near its maximum throughout the mission. In takeoff, the generator has a higher efficiency because of a better position in the operational characteristics and therefore less waste heat than in design. A 50%-water-glycol mixture is chosen as the coolant to cope with the low ambient temperatures at high altitudes.

### 3.1. One-Dimensional Sensitivities

The one-dimensional sensitivity analysis considers the sensitivity of each parameter isolated, i.e., only one parameter is varied at a time. In Section 3.2, some coupled or multidimensional sensitivities are discussed. The parameters considered for the one-dimensional analysis are summarized in Table 2 and the results are shown in Figure 7.

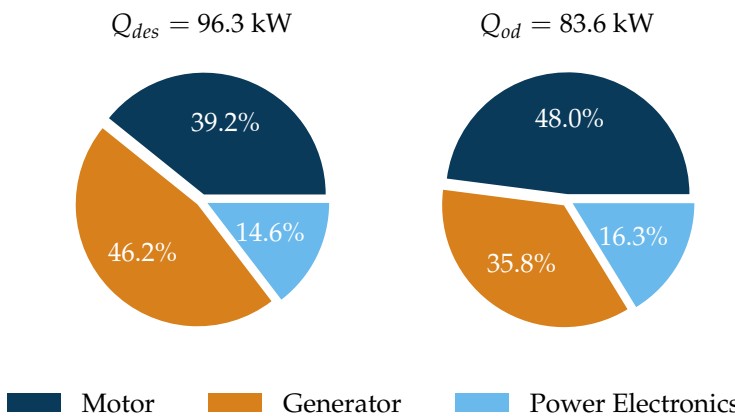

**Figure 6.** Design and off-design (HDTO) heat loads for $S_P = 30\%$ for one powertrain.

**Table 2.** Parameters considered in the one-dimensional sensitivity analysis.

| Parameter | Symbol | Unit | Default Value |
|---|---|---|---|
| Coldplate surface temperature | $T_{cp}$ | K | 370 |
| Heat capacity ratio HEX cold to hot side | $C_R^*$ | – | 1.0 |
| Coldplate coolant inlet temperature | $T_1$ | K | 275 |
| Pressure ratio HEX cold side | $\Pi_c$ | – | 0.95 |
| Hydraulic diameter HEX cold side | $d_{H,c}$ | mm | 10.0 |
| Coldplate effectiveness | $\epsilon_{cp}$ | – | 0.4 |

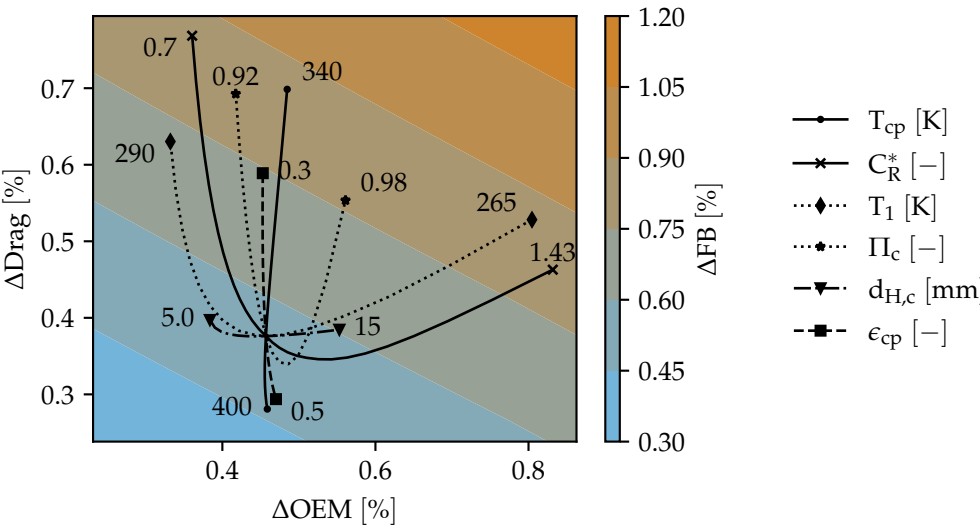

**Figure 7.** One-dimensional sensitivity analysis.

The default values from Table 2 are located at the intersection of all lines in Figure 7. The default values for each parameter are the median values of the respective parameter range. They are mostly not located at the middle of the resulting sensitivity line, indicating a higher sensitivity of the parameter to one end of the range. Increasing $T_{cp}$ by 30 K from 370 K to 400 K, for example, results in roughly a 0.07% decrease in $\Delta FB$, whereas decreasing it by 30 K to 340 K results in an approximate 0.3% increase in $\Delta FB$.

$T_{cp}$ and $\epsilon_{cp}$ have the highest proportionality with $\Delta FB$. Increasing either one of them directly results in an increase of $\Delta T$ across the HEX, which leads to a decrease in HEX size. Both parameters cannot be freely chosen, but $T_{cp}$ is constrained by the allowed operating temperature of the electric component and $\epsilon_{cp}$ by the possible size of the coldplate. High

$\epsilon_{cp}$ values require a longer length of stay of the cooling fluid inside the coldplate, which causes an increase in the size of the coldplate for constant heat loads.

All other parameters have an optimal value with a minimum in $\Delta FB$ inside the given range. Decreasing $C_R^*$ to values lower than 1.0 is a direct increase of $w_c$. This improves the cold-side heat transfer coefficient ($\alpha_c$), which results in a slightly smaller and lighter HEX; however, the corresponding increase in drag leads to an overall increased $\Delta FB$. Increasing $C_R^*$ past 1.0 has the opposite effect. The increased $w_h$ causes an increased hot-side length ($L_h$) of the HEX to achieve the same $T_1$. This allows a shorter cold-side length ($L_c$) and thereby less drag. There is a limit to this effect—it will eventually result in an increase in drag again due to an unnecessarily large HEX area.

$\Pi_c$ has a higher drag than OEM sensitivity. Low $\Pi_c$ values directly result in more drag but also allow slightly lighter systems due to the increased cold-side flow velocity and thus higher $\alpha_c$. The increased drag towards very high $\Pi_c$ values originates in the diffuser. Very low face $Ma$ are required for the HEX, leading to a large diffuser with larger internal losses and also larger spillage.

Decreasing $T_1$ further from the default value requires a more effective HEX, i.e., a larger HEX with increased $L_h$ and $L_c$. Besides becoming heavier, the increased $L_c$ also results in more drag for the system. At constant $\Pi_c$, an increased $L_c$ requires a smaller face $Ma$ with the above-described consequences for the diffuser. However, $T_1$ should not be infinitely increased either. Large $T_1$ values require large $w_h$ values, a constant $C_R^*$, and large $w_c$ values, resulting in a steep increase in drag.

$d_{H,c}$ is inversely proportional to mass because $\alpha_c$ increases with decreasing $d_{H,c}$. A very small $d_{H,c}$ leads to increased $FB$ since for constant $\Pi_c$ a very low face $Ma$ is required, which again causes large diffuser losses and consequently drag, as mentioned above.

### 3.2. Multi-Dimensional Sensitivities

The results of Figure 7 may not be used to observe the optimal value for each parameter. This would only be possible if they were independent of each other. In reality, they are linked to each other via various interdependencies. Some of the more interesting ones are shown in Figure 8. Contrary to Figure 7, the lines in Figure 8 are lines of constant parameter values—e.g., along the dotted lines of the left image, $\Pi_c$ has a constant value, which is indicated on the left side of each line.

The study settings are the same as in Table 2, except for the indicated parameters. On the left side $d_{H,h}$ and $\Pi_c$ are varied. Varying $\Pi_c$ shows the same curve shapes for each $d_{H,h}$ as in Figure 7. The minimum in $\Delta FB$ shifts. For $d_{H,h} = 4.5$ mm, the best $\Pi_c$ would be about 0.94, whereas for smaller $d_{H,h}$, the ideal $\Pi_c$ value increases slightly to about 0.96 for $d_{H,h} = 0.5$ mm. $d_{H,h}$ shows a rather clear trend indicating that lower $d_{H,h}$ values always result in less $\Delta FB$. This statement is only valid as long as $\Pi_c$ can be appropriately chosen. If, for example, $\Pi_c$ is fixed at 0.9, the best $d_{H,h}$ value would be roughly 2 mm.

The main reason for the effects described above is the influence of $d_{H,h}$ on the HEX cold-side ratio of the free flow to the frontal area ($\sigma_c$). A larger $d_{H,h}$ increases the hot-side channel height (if the channel aspect ratio is not changed) and therefore decreases $\sigma_c$. If $\Pi_c$ is left constant, the flow velocity in the cold-side channel is also about constant. However, due to the lower $\sigma_c$ value, the face $Ma$ must be smaller since a lower $\sigma_c$ results in a higher difference between frontal and free flow velocity. Therefore, the diffuser must be larger, resulting in more drag. The system mass always decreases with decreasing $d_{H,h}$ because of an increased $\alpha_h$ and a more compact HEX.

On the right, the effects of varying $d_H$ on both cold and hot sides of the HEX are shown. Again, reducing $d_{H,h}$ results in less $\Delta FB$ in every case. The reason is the same as described above. The optimal $d_{H,c}$ value depends heavily on the chosen $d_{H,h}$. For a large $d_{H,h}$, a larger $d_{H,c}$ should be chosen. A small $d_{H,c}$ with a large $d_{H,h}$ results in small $\sigma_c$ values with its negative effects on drag as described above. For the lowest considered $d_{H,h}$ of 0.5 mm, the best $d_{H,c}$ value is about 5 mm. The factor between $d_{H,h}$ and the corresponding best $d_{H,c}$ value varies between 2 and 10. This large difference can be attributed to the

different fluid properties, especially the large difference in thermal conductivity between water and air.

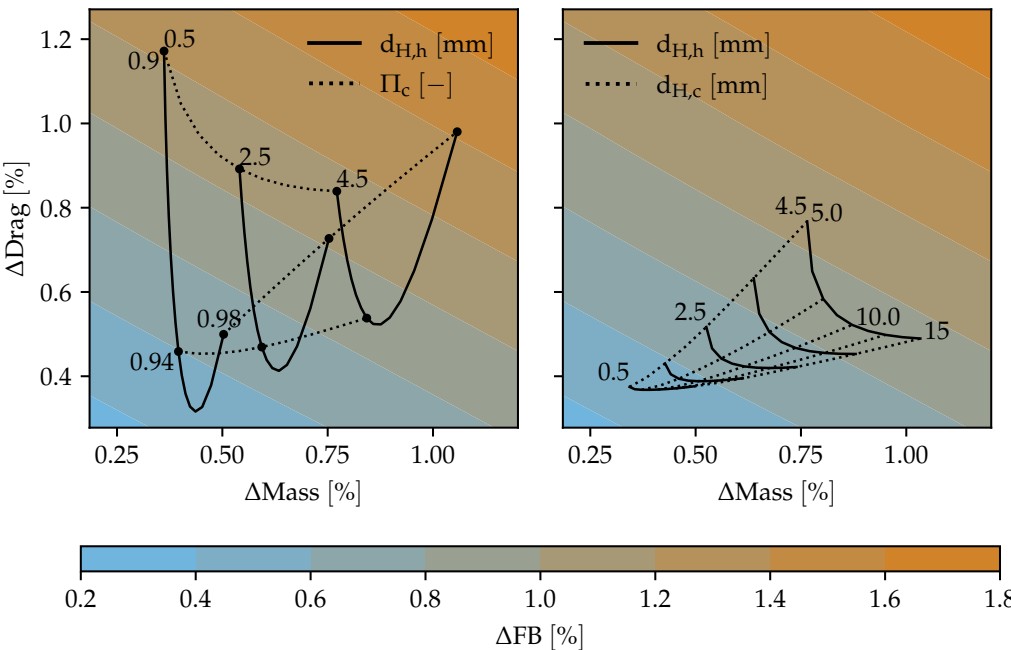

**Figure 8.** Two-dimensional sensitivity analysis of the hot-side hydraulic diameter with the cold-side pressure ratio (**left**) and the hot-side hydraulic diameter with the cold-side hydraulic diameter (**right**).

For TMS-equipped aircraft, a few interesting conclusions can be derived. The general trend in HEX design towards smaller $d_H$ is only beneficial for the aircraft on the hot side if the drag is considered. Studies only focusing on HEX masses will still find smaller $d_{H,c}$ beneficial. For practical reasons, $d_{H,h}$ can be reduced far easier than $d_{H,c}$. The smaller the $d_H$, the higher the risk of congestion, and the greater the drop in performance for the HEX. The hot side is a closed loop, and therefore the fluid can be kept very pure through regular exchange and the incorporation of filter systems, thereby minimizing said risk. On the cold side, ambient air has to be used. The implementation of a filter would directly result in more drag and is therefore not a feasible option. With optimal $d_{H,c}$, values of more than 5 mm for maintenance are less of a problem than $d_{H,c}$ values of only a millimeter or less. Due to its obvious trends, $d_{H,h}$ does not need to be considered as a free variable but rather as direct input constrained mainly by manufacturing techniques for the optimization studies in Section 4 if mass, drag, and *FB* are the only relevant metrics.

### 3.3. Heat Exchanger Size

While mass, drag, and *FB* are the most relevant metrics for the aircraft performance, the system size cannot be neglected since the TMS has to be integrated into the aircraft. The influences of $d_{H,h}$ and $d_{H,c}$ on the three HEX dimensions $L_h$, $L_c$, and stack height ($H_{stack}$) are shown in Figure 9. The study settings are equal to those in Figure 8, except for a smaller range of considered values for both $d_H$.

Clearly, $d_H$ on both sides has a direct influence on overall HEX dimensions. In any size constrained optimization problem, $d_{H,h}$ should therefore be considered as a free variable as well. Increasing $d_{H,h}$ results in an increase of $L_h$ because $\Pi_h$ is kept constant. To have the same pressure drop for a lower $f_h$, $L_h$ needs to be higher. As a consequence of the increased $L_h$, $H_{stack}$ is reduced because $Q$ is also constant. Without a reduction in $H_{stack}$, the total heat exchange area would be larger, and therefore $Q$ would be higher than actually required. Increasing $d_{H,c}$ shows an analogue trend.

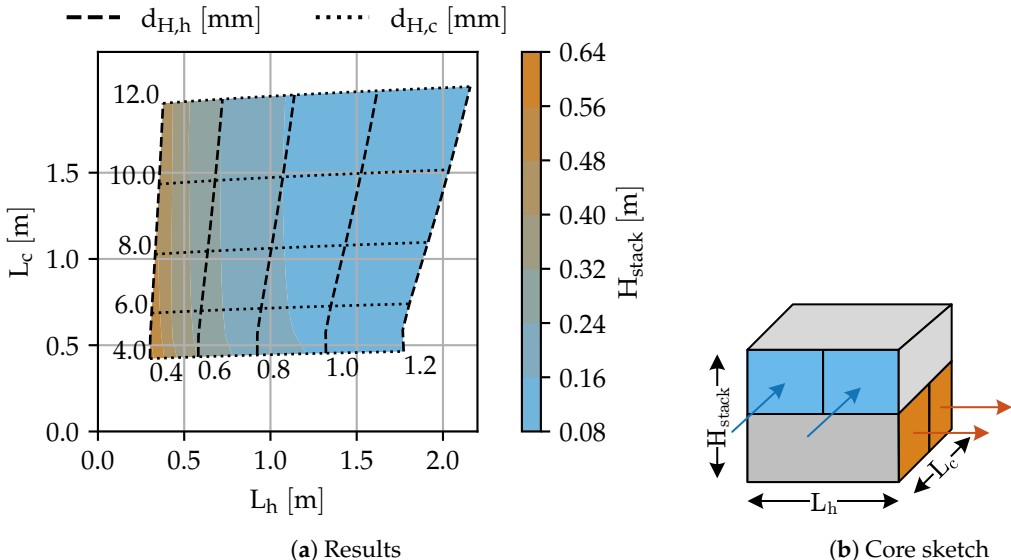

(**a**) Results             (**b**) Core sketch

**Figure 9.** Heat exchanger size sensitivity in three dimensions: hot-side length, cold-side length, and stack height over hot- and cold-side hydraulic diameter.

## 4. Design and Off-Design Optimization for the Application Case

This section uses the previously gathered knowledge to design and optimize TMS for the application case of a HEA (here, "hybrid" refers to power hybridization) described in Section 2.5. The section is divided into design, off-design, and multi-point design.

### 4.1. Design Point Optimization

The settings of the study have already been described in the previous sections. The design point of choice is the TOC, which is also the design point of the gas turbine. The aircraft has been designed with three different $S_P$ values, so a TMS was designed for each of them. Free variables for the optimization were $d_{H,h}$, $d_{H,c}$, $\Pi_h$, $\Pi_c$, $C_R$, $(A_0/A_1)_{diff}$, and $T_1$. The cumulative optimization results of two identical TMS (one for each powertrain) are shown in Figure 10.

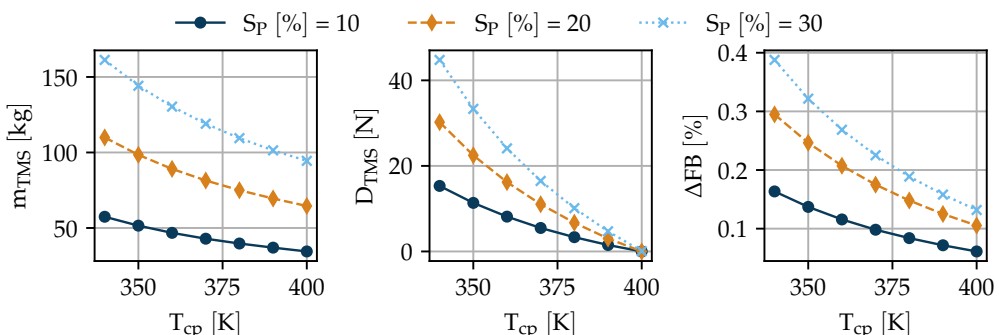

**Figure 10.** Design optimization results for different $S_P$ variations.

For each $S_P$, multiple designs for different $T_{cp}$ were made, as $T_{cp}$ is subject to electric component technology and therefore not certainly known. $S_P$ values were imposed by the aircraft studies [26], and the given range of $T_{cp}$ was chosen to include current electric component technology. Results are shown for $m_{TMS}$, $D_{TMS}$, and $\Delta FB$, which was the objective function of the optimization. As expected, all three parameters grow with increasing $S_P$ and decreasing $T_{cp}$. The exponential behavior towards decreasing $T_{cp}$ was also anticipated from the results shown in Figure 7. $D_{TMS}$ gets reduced to almost 0 N when increasing $T_{cp}$ to 400 K, due to the Meridith effect. The heat rejected by the HEX is recovered as thrust and compensates for the pressure loss of the TMS. If even higher $T_{cp}$ values are possible,

the aircraft *FB* sensitivities have to be extended towards negative drags, i.e., thrust from the TMS.

With state-of-the-art electric components, i.e., motors, generators, and power electronics, a $T_{cp}$ of 380 K is realistic. For the three different $S_P$ values, $\Delta FB$ is 0.09%, 0.15%, and 0.19%, respectively. There are several reasons for these very low values. Firstly, $S_P$ is not very large, and therefore $Q$ stays relatively low (see Figure 6). Secondly, the partial-turboelectric architecture only includes electric components with comparably high maximum operating temperatures. If a large battery or fuel cell is included in the powertrain, the TMS design becomes more complex and will likely have a higher impact on $\Delta FB$. Thirdly, the currently implemented system mass estimations have to be refined in a more detailed analysis. So far, redundancy is not considered. Additionally, the technology assumptions for the HEX have been rather optimistic, with wall thicknesses for the plates assumed at 0.5 mm and for the fins at 0.1 mm.

Fourthly, integration of the TMS has not been considered in the design yet. For $T_{cp}$ = 380 K and $S_P$ = 30%, the HEX would measure $L_c \times L_h \times H_{stack}$ = 0.48 m × 0.73 m × 0.18 m. The diffuser and nozzle would be 2.2 m and 0.9 m long, respectively, resulting in an overall cold-side system length of 3.5 m. If needed, the diffuser could be shortened, trading efficiency. The current model (see Section 2.3) only allows diffusers with $\theta = 8°$. In this case, a fuselage integration seems feasible, but cargo space would be reduced. Another option could be the installation on top of the wing near the root, but it would possibly require additional cowlings, resulting in additional mass and drag.

It is worth noting that the numeric optimization resulted in $\Delta FB$ values of less than 0.2% for $S_P$ = 30% and $T_{cp}$ = 380 K, whereas even the best values in the sensitivity studies (see Figures 7 and 8) were above 0.4%. While the difference in percentage points is not of large relevance to the aircraft in this case, the relative difference achieved through numeric optimization is remarkable, i.e., a reduction of more than 50%.

### 4.2. Off-Design Point Optimization

An exemplary off-design optimization was conducted for $T_{cp,des}$ = 380 K and $S_P$ = 30%. The objective function was the electric power required to drive the TMS ($P_{TMS}$), which includes the power for the hydraulic pump and the fan. The efficiencies of the pump $\eta_{hyd}$ and $\eta_{elec}$ were assumed to be 0.75 and 0.95, respectively, and the fan efficiency ($\eta_{fan}$) was set to 0.50. In a more detailed study, proper maps should be implemented for pump and fan efficiency to accurately predict their behavior with changing operating conditions. Variables of the study were the international standard atmosphere (ISA) temperature deviation ($\Delta T_{ISA}$) and the differences between cooling fluid outlet and inlet temperatures of the electric components ($\Delta T_{cp}$). The results are shown in Figure 11.

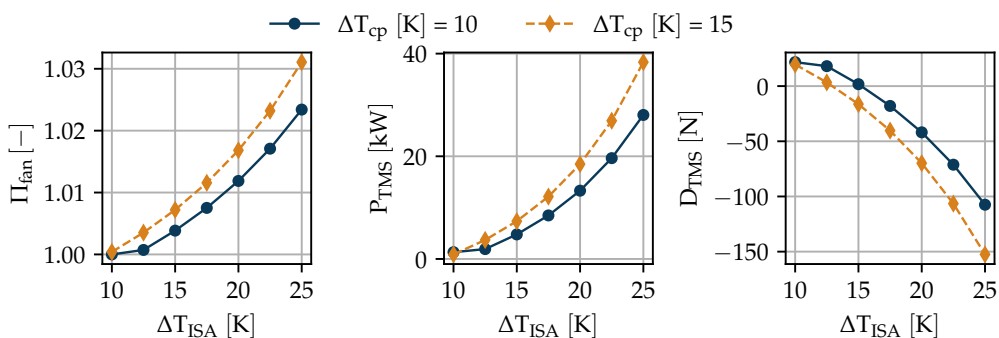

**Figure 11.** Off-design optimization at takeoff for a TMS designed for $T_{cp,des}$ = 380 K and $S_P$ = 30%.

Hot days are a particular challenge for the TMS because the available $\Delta T$ between cooling fluid and ambient is smaller. Raising $\Delta T_{ISA}$ results in an exponential increase in required fan pressure ratio ($\Pi_{fan}$). $\Delta T_{cp}$ is an operational parameter that can be controlled via $P_{pump}$. A lower $P_{pump}$ results in a smaller $w_h$, and thereby a higher $\Delta T_{cp}$. A higher $\Delta T_{cp}$

value does require a larger $\Pi_{fan}$ because the $\Delta T$ between hot-side HEX inlet to outlet is larger, and therefore a higher $\alpha_c$ is needed. $P_{TMS}$ follows $\Pi_{fan}$ almost directly because $P_{pump}$ is at a different order of magnitude, i.e., only 1.1 kW and 0.5 kW for $\Delta T_{cp} = 10$ K and 15 K, respectively. The large difference between $P_{pump}$ and $P_{fan}$ is due to the fact that the pump compresses an incompressible fluid, and the compressor a compressible one. About 25% $P_{TMS}$ can be saved on a hot day by choosing the lower $\Delta T_{cp}$ value.

$P_{TMS}$ has not been considered in the aircraft *FB* sensitivities. During the majority of the mission, the fan is not required and could either be removed from the flow path or set to idle. The takeoff segment is rather short compared to the overall mission length, and even if the maximum load of 60 kW is required, the impact on the powertrain is negligible. The generators have a combined power of more than 2 MW, and some of the $P_{TMS}$ is actually converted to useful thrust, as seen by the negative drag values of up to $-150$ N.

### 4.3. Multi-Point Optimization

From the previous section, the question arises of whether it is possible to design a TMS without the additional puller fan. Though its impact on $\Delta FB$ is negligible, it is still an additional component with costs and requirements for certification and maintenance. To answer the question, a multi-point study was conducted that combined the previous design point with an additional constraint to achieve the required cooling power in off-design as well. The objective function was again $\Delta FB$—the same as in Section 4.1. The results are shown in Figure 12.

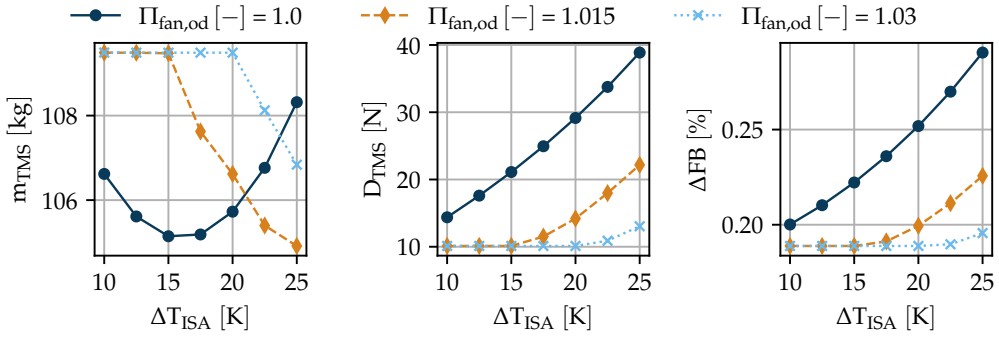

**Figure 12.** Multi-point optimization for a TMS for $T_{cp,des} = 380$ K and $S_P = 30\%$.

Three different off-design fan pressure ratios ($\Pi_{fan,od}$) were investigated. If $\Pi_{fan,od}$ is 1.0, no fan installation is required. For the larger values of $\Pi_{fan,od}$, all results form horizontal lines for lower $\Delta T_{ISA}$. This implies that the optimal design is only dependent on the design point, and the additional off-design constraint is met because $\Pi_{fan,od}$ is oversized. Only when $\Delta T_{ISA}$ increases beyond a certain threshold, the off-design constraint becomes relevant.

If no fan is installed ($\Pi_{fan,od} = 1.0$), the constraint is relevant even at low $\Delta T_{ISA}$, immediately resulting in a larger TMS with increased $\Delta FB$. $\Delta FB$ grows exponentially with $\Delta T_{ISA}$. It is certainly possible to design the TMS without the puller fan, however, assuming a maximum $\Delta T_{ISA}$ of 25 K, $\Delta FB$ would increase from 0.19% to 0.29%. In absolute numbers, this difference is negligible, but for a TMS with a larger *FB* impact, it could be better to install the fan. Using a puller fan also has the advantage of an additional degree of freedom for the system that can help to better adapt to operational changes.

## 5. Conclusions and Outlook

Ram air–based thermal management systems (TMS) were investigated regarding their overall impacts on an aircraft's fuel burn. Fuel burn sensitivities were derived from a 180-passenger short-range partial-turboelectric aircraft equipped with wingtip propellers by adding an assumed TMS design drag and mass to it.

A TMS model consisting of coldplates for heat acquisition, pipes and pumps for hot-side heat transfer, a two-pass cross-flow plate-fin heat exchanger for heat rejection, and a diffuser and a nozzle for cold-side flow velocity control was developed. Variations of one- and multi-dimensional parameter sensitivities were used to gain an understanding of the system. The system reacted very sensitively to seven parameters that were selected as free variables for a numeric optimization.

Alternating the hydraulic diameter of the main heat exchanger on both sides was shown to be one of the most effective ways to control the overall system dimensions and therefore manage the integration problem.

TMS optimization studies were conducted. It was found that increasing electric component junction temperature to about 400 K could eliminate parasitic drag from the TMS in cruise entirely. For a more realistic temperature of 380 K, additional fuel burn for an aircraft with 30% power split was 0.19%. The system could withstand hot-day takeoff conditions with the help of a small puller fan installed behind the main heat exchanger. Alternatively, oversizing the TMS removed the need for a puller fan but increased additional fuel burn to 0.29%.

In the future, the mass of the system should be re-investigated. Redundancy considerations are most likely going to cause an increase in system mass of up to 100%. In this study, only rectangular channels were considered for the heat exchanger core. Other options, such as offset-strip fins and louvered fins should be considered in the future. Additionally, integration of the TMS, including secondary mass and drag increases, will be discussed in the future. The integration of the TMS seems to be one of the largest challenges. In concrete aircraft applications, this problem should be addressed and possibly solved in a synergistic manner—e.g., by installing the ram-air inlets behind an open rotor. Additionally, adaptive nozzle geometries are an idea to better adapt TMS performance in different operating conditions.

**Author Contributions:** Conceptualization, methodology, simulation, analysis, and writing of all aspects of the research except for Section 2.5, H.K.; conceptualization, methodology, simulation, analysis, and writing of all aspects of the research of Section 2.5, M.L.; conceptualization, methodology, simulation, and analysis of the powertrain, M.P.; supervision, M.H. All authors have read and agreed to the published version of the manuscript.

**Funding:** This research received funding as part of the IVeA project, a research project supported by the Federal Ministry for Economic Affairs and Energy in the national LuFo program.

**Data Availability Statement:** Not applicable.

**Acknowledgments:** We would like to thank everyone involved in the IVeA project for their dedication to its success. Additionally, we thank Arne Seitz for his continued support and fruitful discussions.

**Conflicts of Interest:** The authors declare no conflict of interest. The funders had no role in the design of the study; in the collection, analyses, or interpretation of data; in the writing of the manuscript, or in the decision to publish the results.

## Abbreviations

| | |
|---|---|
| HDTO | Hot-day takeoff |
| HEA | Hybrid-electric aircraft |
| HEX | Heat exchanger |
| ISA | International standard atmosphere |
| MP | Main propeller |
| NASA | National Aeronautics and Space Administration |
| OEM | Operating empty mass |
| PCHE | Printed circuit heat exchanger |
| PFHE | Plate fin heat exchanger |
| TMS | Thermal management system |

| TOC | Top of climb | |
| VTOL | Vertical takeoff and landing | |
| WTP | Wingtip propeller | |

**Roman Symbols**

| $A$ | Area | m$^2$ |
| --- | --- | --- |
| $A_R$ | Diffuser area ratio | $-$ |
| $\overline{A_R}$ | Corrected diffuser area ratio | $-$ |
| $b$ | Heat exchanger plate space | m |
| $B$ | Diffuser inlet blockage | $-$ |
| $c_p$ | Specific heat capacity at constant pressure | J/(kgK) |
| $c_p^*$ | Ideal diffuser pressure recovery factor | $-$ |
| $c_v$ | Specific heat capacity at constant volume | J/(kgK) |
| $C$ | Absolute heat capacity | W/K |
| $C_R$ | Heat capacity ratio ($C_{min}/C_{max}$) | $-$ |
| $C_R^*$ | Side-specific heat capacity ratio ($C_h/C_c$) | $-$ |
| $cmv$ | Core mass velocity | kg/(m$^2$s) |
| $d_H$ | Hydraulic diameter | m |
| $D$ | Drag | N |
| $f$ | Fanning friction factor | $-$ |
| $FB$ | Fuel burn | kg |
| $g$ | Diffuser pressure recovery geometry factor | $-$ |
| $j$ | Colburn factor | $-$ |
| $K_{bt}$ | Bend loss coefficient | $-$ |
| $K_c$ | Inlet loss coefficient | $-$ |
| $K_e$ | Outlet loss coefficient | $-$ |
| $K_{loss}$ | Nozzle pressure loss coefficient | $-$ |
| $K_{spill}$ | Spillage coefficient | $-$ |
| $L$ | Length | m |
| $m$ | Mass | kg |
| $Ma$ | Mach number | $-$ |
| $n_p$ | Number of passes | $-$ |
| $ntu$ | Number of transfer units on one side | $-$ |
| $NTU$ | Number of transfer units | $-$ |
| $p$ | Pressure | Pa |
| $P$ | Power | W |
| $Pr$ | Prandtl number | $-$ |
| $q$ | Area-specific heat flow rate | W/m$^2$ |
| $Q$ | Heat flow rate | W |
| $r_{th}$ | Thermal insulance | m$^2$K/W |
| $R_{th}$ | Thermal resistance | K/W |
| $Re$ | Reynolds number | $-$ |
| $S_P$ | Power split | % |
| $t$ | Channel width | m |
| $T$ | Temperature | K |
| $U$ | Overall heat transfer coefficient | W/(m$^2$K) |
| $v$ | Velocity | m/s |
| $V$ | Volume | m$^3$ |
| $w$ | Mass flow rate | kg/s |

**Greek Symbols**

| $\alpha$ | Heat transfer coefficient | W/(m$^2$K) |
| --- | --- | --- |
| $\delta$ | Fin thickness | m |
| $\Delta$ | Difference | $-$ |

| | | |
|---|---|---|
| $\epsilon$ | Heat exchanger effectiveness | $-$ |
| $\eta_o$ | Overall fin efficiency | $-$ |
| $\Phi$ | Aspect ratio | $-$ |
| $\Pi$ | Pressure ratio | $-$ |
| $\rho$ | Density | kg/m$^3$ |
| $\rho_A$ | Area density | kg/m$^2$ |
| $\sigma$ | Heat exchanger ratio of free flow to frontal area | $-$ |
| $\theta$ | Diffuser opening angle | deg |

**Subscripts**

| | |
|---|---|
| *c* | Cold |
| *cond* | Conductive |
| *conv* | Convective |
| *corr* | Corrected |
| *cp* | Coldplate |
| *cs* | Cross section |
| *des* | Design |
| *f* | Finned |
| *h* | Hot |
| *i* | Inlet |
| *m* | Mean |
| *o* | Outlet |
| *od* | Off-design |
| *s* | Static |
| *spill* | Spillage |
| *tot* | Total |

## Appendix A. Coldplate Model

*Appendix A.1. Model Description*

For the coldplate design model, all input and output parameters are listed in Table A1. The input parameters have to be estimated or obtained from manufacturer data.

The area-specific heat load ($q_{des}$) is calculated from the thermal insulance ($r_{th,des}$) and the coldplate surface temperature ($T_{cp,des}$) [27,28].

$$q_{des} = (T_{cp,des} - T_{i,des})/r_{th} \tag{A1}$$

The outlet temperature ($T_{o,des}$) can be obtained from the effectiveness ($\epsilon_{des}$). The evaluation of fluid properties inside a heat exchanging device is conducted at an average temperature ($T_m$):

$$T_{o,des} = (T_{cp,des} - T_{i,des}) \cdot \epsilon_{des} + T_{i,des} \tag{A2}$$

$$T_m = (T_i + T_o)/2 \tag{A3}$$

The specific heat capacity of the cooling fluid ($c_v$) is a function of $T_m$ and $p_i$ (the pressure drop is neglected here as $c_v$ has a much larger temperature than pressure sensitivity) and is evaluated from the CoolProp fluid database [29]. The required mass flow ($w_{des}$) can be calculated from $Q_{des}$ and the area of the coldplate ($A_{cp}$) from $q_{des}$:

$$w_{des} = Q_{des}/(c_v \cdot (T_{o,des} - T_{i,des})) \tag{A4}$$

$$A_{cp} = Q_{des}/q_{des} \tag{A5}$$

The dry mass is then calculated from the area density ($\rho_A$):

$$m_{dry} = A_{cp} \cdot \rho_A \tag{A6}$$

The product of the overall heat transfer coefficient and heat exchange area ($(UA)_{des}$) is required for later off-design calculations (note: $A_{des} \neq A_{cp}$ since $A_{cp}$ is the coldplate base area and $A_{des}$ the inner channel surface area). It is calculated from the number of transfer units ($NTU$). The $NTU$–$\epsilon$ relation for heat exchanging devices with a heat capacity ratio of $C_r = 0$ is found in many thermodynamic textbooks, e.g., [24].

$$NTU_{des} = -\ln(1 - \epsilon_{des}) \tag{A7}$$

$$UA_{des} = NTU_{des} \cdot c_p \cdot w_{des}/A_{cp} \tag{A8}$$

Finally, the outflow pressure ($p_o$) is calculated:

$$p_{o,des} = p_{i,des} - \Delta p_{des} \tag{A9}$$

**Table A1.** Design parameters for the coldplate model.

| Parameter | Symbol | Unit |
|---|:---:|:---:|
| Inputs | | |
| Inlet pressure | $p_{i,des}$ | Pa |
| Inlet temperature | $T_{i,des}$ | K |
| Effectiveness | $\epsilon_{des}$ | − |
| Heat load | $Q_{des}$ | W |
| Coldplate surface temperature | $T_{cp,des}$ | K |
| Thermal insulance | $r_{th,des}$ | $m^2K/W$ |
| Area density | $\rho_A$ | $kg/m^2$ |
| Pressure drop | $\Delta p_{des}$ | Pa |
| Outputs | | |
| Design mass flow | $w_{des}$ | kg/s |
| Outlet pressure | $p_{o,des}$ | Pa |
| Outlet temperature | $T_{o,des}$ | K |
| Area-specific heat load | $q_{des}$ | $W/m^2$ |
| Coldplate area | $A_{cp}$ | $m^2$ |
| Dry mass | $m_{dry}$ | kg |
| Number of transfer units | $NTU_{des}$ | − |
| U-A product | $(UA)_{des}$ | W/K |

In off-design calculations, the dimensions of the coldplate are fixed. Only fluid inlet conditions ($T_i$, $p_i$, $w_{od}$) vary, as does the off-design heat load ($Q_{od}$). All input and output parameters of the off-design model are listed in Table A2.

Since $A_{cp}$ has been defined in the design model, the off-design area-specific heat flow ($q_{od}$) can be calculated:

$$q_{od} = Q_{od}/A_{cp} \tag{A10}$$

$T_m$ is calculated from (A3), and $c_p$ is obtained from tabulated data. The off-design mass flow ($w_{od}$) is determined from (A4) with off-design inputs. The off-design coldplate temperature ($T_{cp,od}$) can be determined from the off-design effectiveness ($\epsilon_{od}$).

$$NTU_{od} = (UA)_{od}/(c_p \cdot w_{od}) \tag{A11}$$

$$\epsilon_{od} = 1 - e^{-NTU_{od}} \tag{A12}$$

$$T_{cp,od} = q_{od}/h_{od} + T_m \tag{A13}$$

With $(UA)_{od} = (UA)_{des}$. This will be proven in the following paragraph.

**Table A2.** Off-design parameters for the coldplate model.

| Parameter | Symbol | Unit |
|---|---|---|
| Inputs | | |
| Inlet pressure | $p_i$ | Pa |
| Inlet temperature | $T_i$ | K |
| Outlet temperature | $T_o$ | K |
| Heat load | $Q_{od}$ | W |
| Outputs | | |
| Off-design mass flow | $w_{od}$ | kg/s |
| Outlet pressure | $p_o$ | Pa |
| Coldplate temperature | $T_{cp,od}$ | K |
| Area-specific thermal resistance | $r_{th,od}$ | m$^2$K/W |
| Effectiveness | $\epsilon_{od}$ | − |

The area is constant as no geometries are changed. For a coldplate, $U$ is comprised of conductive ($\alpha_{cond}$) and convective ($\alpha_{conv}$) heat transfer coefficients. $\alpha_{cond}$ does not change in off-design situations because material and thickness are constant. The change in thermal conductivity of the material ($\lambda$) is neglected because the mean material temperature is not expected to differ greatly between design and off-design. $\alpha_{conv}$ can be calculated from:

$$\alpha_{conv} = Nu \cdot \lambda / d_H \tag{A14}$$

with Nusselt number ($Nu$) and hydraulic diameter ($d_H$) [30]. Microchannels provide compact, light-weight coldplates with the ability to absorb very high $q_{des}$ as required by modern chip generations. The flow in such small channels is typically laminar due to the very small $d_H$ [31]. In laminar flow, $Nu$ is constant regardless of the flow velocity [24]. For this model, laminar flow is assumed in all operating points. To ensure this assumption is true, the coldplate should always be designed for maximum mass flow, and off-design operating points should have smaller mass flows ($w_{des} > w_{od}$). $d_H$ is also constant as it is a fixed geometry. Neglecting the $T$-$p$ dependency of $\lambda$ the equity of both $\alpha_{conv}$ follows:

$$\alpha_{conv,des} = \alpha_{conv,od} \tag{A15}$$

$U_{des}$ is known from (A7). If the temperature differences between design and off-design are large, the $\lambda$-$T$-$p$ sensitivity can be accounted for by means of a ratio $\lambda_{od}/\lambda_{des}$. The off-design thermal insulance ($r_{th,od}$) is:

$$r_{th,od} = (T_{cp,od} - T_i) / q_{od} \tag{A16}$$

Since no exact geometry is known from the design model, the off-design pressure loss ($\Delta p_{od}$) has to be derived from its design counterpart and the design/off-design $w$-ratio:

$$\Delta p_{od} = f(\Delta p_{des}, w_{des} / w_{od}) \tag{A17}$$

In general, $\Delta p$ can be calculated from [24]:

$$\Delta p = h \cdot \rho \cdot g \tag{A18}$$

with head loss ($h$) and gravitational constant ($g$). The head loss is [32]:

$$h = f \cdot L \cdot u^2 / (d_H \cdot 2 \cdot g) \tag{A19}$$

with friction factor ($f$), flow length ($L$), and flow velocity ($u$). In laminar flow, $f$ is a function of $Re$ and a channel geometry depending constant ($c_{geom}$) [24]:

$$f = c_{geom}/Re \tag{A20}$$

$$Re = u \cdot d_H/\nu \tag{A21}$$

with kinematic viscosity ($\nu$). Combining (A18)–(A21) results in:

$$\Delta p = c_{geom} \cdot L \cdot u \cdot \rho \cdot \nu/(2 \cdot d_H^2) \tag{A22}$$

$c_{geom}$, $L$, and $d_H$ do not change from design to off-design conditions; the difference in $\rho$ and $\nu$ is neglected so that:

$$\Delta p_{od}/\Delta p_{des} = u_{od}/u_{des} \tag{A23}$$

$$u = w/(\rho \cdot A_{cs}) \tag{A24}$$

with flow cross section area ($A_{cs}$). Again, $A_{cs}$ stays constant, and the difference in $\rho$ is neglected, finally resulting in:

$$\Delta p_{od} = \Delta p_{des} \cdot w_{od}/w_{des} \tag{A25}$$

$p_{o,od}$ can now be calculated via (A9).

*Appendix A.2. Coldplate Validation Design Inputs*

**Table A3.** Design inputs for coldplate validation.

| Parameter | Unit | Value |
|---|---|---|
| $T_{1,des}$ | K | 294 |
| $\epsilon_{des}$ | — | 0.47 |
| $Q_{des}$ | W | 100 |
| $T_{cp,des}$ | K | 330 |
| $r_{th,des}$ | m$^2$K/W | $2.88 \times 10^{-5}$ |
| $\Delta p_{des}$ | Pa | $50 \times 10^3$ |

**Appendix B. Compact Heat Exchanger Core Model**

This section describes how the core geometry parameters and Colburn factor ($j$) and Fanning friction factor ($f$) for the different core surfaces of a compact heat exchanger are calculated.

1. **Rectangular microchannels**. $j$ and $f$ are calculated according to the methods described for rectangular channels in [24]. Of the parameters in Table 1, $d_H$ and $\delta$ are used as known inputs, and the other parameters are calculated. The aspect ratio of the channels is also an input and defined as:

$$\Phi = b/t \tag{A26}$$

with channel width ($t$). Starting from (A26) and the definition of $d_H$:

$$d_H = \frac{4A_{cs}}{P} \tag{A27}$$

with channel cross section area $A_{cs}$ and perimeter $P$, rearrangement leads to:

$$b = d_H \cdot \frac{1 + \Phi}{2} \tag{A28}$$

In a similar fashion, using basic geometry and regarding the sidewalls of the channels as fins results in:

$$A_f/A = \frac{\Phi}{\Phi + 1} \tag{A29}$$

with finned area $A_f$ and total heat exchange area $A$. The area density is defined as:

$$\beta = \frac{A}{V} \tag{A30}$$

with core volume $V$. Combining (A26), (A27), and (A30) concludes after some rearrangements in:

$$\beta = \frac{4 \cdot (1 + \Phi)}{d_H \cdot (1 + \Phi) + 2 \cdot \Phi \cdot \delta} \tag{A31}$$

2. **Offset-strip fins**. The model for this core is entirely based on [33]. $j$ and $f$ correlations were directly adapted and used within the given limits. For offset-strip fins, the fin length ($L_f$) is required as an additional input parameter. The missing geometries were derived from Figure 1 in [33]. If offset-strip fins could be realized without additional material on the top or bottom $b$, $A_f/A$, and $\beta$ could be calculated from (A28), (A29), and (A30), respectively. With enhanced manufacturing techniques, it may become possible. Hence, for this model, the additional material thickness on the top and bottom is neglected.

3. **Louvered fins**. The correlation for $j$ was directly implemented from [34] and for $f$ from [35]. $b$ is used as a direct input for this model. $A_f/A$ and $\beta$ were calculated with (8.76–8.84) from [17]. Additional input parameters to be considered here are louver angle, louver pitch, and louver cut length, which should be selected carefully within the valid ranges given in [34,35].

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
