# Peer review of "Design and Optimization of Ram Air–Based Thermal Management Systems for Hybrid-Electric Aircraft"

_aerospace, doi:10.3390/aerospace8010003_

Round 1

Reviewer 1 Report

The article Design and Optimization of Ram Air-Based Thermal

Management Systems for Hybrid-Electric Aircraft may be accepted after it solves some minor issues.

  • Some acronyms need to be defined before to be used (i.e. FB)
  • Do the pump an the fan efficienc are considered constant for any conditions?
  • Figures 4 and 5 do not seem to be needed. It is not clear what is the information that they provide.
  • In Figure 6, indicates which refers to design and off design conditions. The off design condition does not change. Is it a binary situation?.
  • It is needed to justify the range used in the optimization.

Author Response

Dear reviewer,

thank you very much for the very positive review of our work. We tried to address your comments as best as possible. In detail:

We added a declaration of the acronym "FB" in the text (section 2.5) and scanned the entire document for other missing acronym declarations.

Yes, pump and fan efficiencies are considered constant throughout our studies. We added some information on the numerical values in section 4.2. We found this the best-suited location as Figure 11 includes the power required by the system. We also critically addressed the simplicity of these assumptions and recommend the use of efficiency maps for future, more detailed studies.

We agree that Figures 4 and 5 may not have as much or as direct information as most of the other figures. However, we feel that they aid in understanding the aircraft application. Figure 4 by visualizing the aircraft and thereby quickly giving the reader an impression of the aircraft. Figure 5 is a result of aircraft sensitivity studies and allows the reader to grasp the dimension of the expected fuel burn change due to the TMS. Therefore, we would prefer to include the graphics if possible, even if they are only nice to have rather than necessary.

Yes, you could consider the operating points binary. We tried to address the most critical off-design point in this study. For a more detailed analysis, a full mission profile should be used. We added clarification of the exact off-design condition (Hot day take-off) in the caption of Figure 6.

We were not entirely sure, what exactly you meant by "range of the optimization". Guessing that you meant the parameters "S_P" and "T_cp" from Figure 10, we included some information of why these ranges were chosen in section 4.1.

Please consult the review cover letter for the exact line numbers of the implemented changes.

We very much hope that our revision is to your satisfaction. Please let us know if you request any further changes.

Sincerely,

Hagen Kellermann, Michael Lüdemann, Markus Pohl, Mirko Hornung

Reviewer 2 Report

This is a very well-written manuscript on an interesting topic, and I am happy to recommend publication in Aerospace.  I especially appreciate the list of acronyms at the end, which is most helpful to the reader.

Author Response

Dear reviewer,

Thank you very much for the very positive feedback on our work. It is greatly appreciated.

Sincerely,

Hagen Kellermann, Michael Lüdemann, Markus Pohl, Mirko Hornung